# Mechanism for Stabilizing an Amorphous Drug Using Amino Acids within Co-Amorphous Blends

**DOI:** 10.3390/pharmaceutics15020337

**Published:** 2023-01-19

**Authors:** Yannick Guinet, Laurent Paccou, Alain Hédoux

**Affiliations:** UMR 8207, UMET—Unité Matériaux et Transformations, University Lille, CNRS, INRAE, Centrale Lille, F-59000 Lille, France

**Keywords:** Raman spectroscopy, glass transition, hydrogen bond, physical stability, cryo-milling

## Abstract

Designing co-amorphous formulations is now recognized as a relevant strategy for improving the bioavailability of low-molecular-weight drugs. In order to determine the most suitable low-molecular-weight excipients for stabilizing the drug in the amorphous state, screening methods were developed mostly using amino acids as co-formers. The present study focused on the analysis of the thermal stability of co-amorphous blends prepared by cryo-milling indomethacin with several amino acids in order to understand the stabilization mechanism of the drug in the amorphous state. Combining low- and mid-frequency Raman investigations has provided information on the relation between the physical properties of the blends and those of the H-bond network of the amorphous drug. This study revealed the surprising capabilities of L-arginine to stiffen the H-bond network in amorphous indomethacin and to drastically improve the stability of its amorphous state. As a consequence, this study suggests that amino acids can be considered as stiffeners of the H-bond network of indomethacin, thereby improving the stability of the amorphous state.

## 1. Introduction

It is well recognized that improving the bioavailability of low-molecular-weight drug candidates is a major concern of pharmaceutical companies, since the majority of small active molecules are synthesized in the crystalline state with inherent poor water solubility [1,2]. Most strategies to overcome this issue are based on designing multi-component formulations which can be roughly divided in two categories. The first category can be formed by systems designed by associating polymers with active pharmaceutical ingredients (APIs), while the second would be composed of two low-molecular-weight co-formers. This second case offers a wide spectrum of strategies, making it possible to design co-crystalline or co-amorphous formulations, and also to associate an excipient with an API or two APIs. Additionally, some multi-component systems can be obtained in the crystalline state (co-crystals) or in the amorphous state (co-amorphous systems), depending on the preparation method used, with possible transformation between amorphous and crystalline states. A lot of co-amorphous formulations are designed using amino acids (AAs) as a co-former and prepared by co-milling [3,4] and other methods [5]. Screening methods have been developed as a strategy for co-former selection to design co-amorphous formations. However, only a few studies focus on the physical mechanisms responsible for the formation of co-amorphous systems which are closely involved in their physical stability [6,7].

The present study aims to analyze multi-component systems producing co-amorphous formulations or co-crystals by co-milling with possible transformation between the two systems. Co-amorphous formulations were prepared using various AAs, i.e., L-arginine, and three isomers, namely L-Leucine (LEU), L-norleucine (NLE) and L-tert-leucine (TLE), while indomethacin (IMC) was selected as the model drug. The originality of this study results from the use of a single technique, Raman spectroscopy, for analyzing the physical state of the formulations, the phase transformations (glass transition, crystallization) and the intermolecular interactions via H-bonding. This type of study combining low- and high-frequency Raman investigations provides rich and direct information on the influence of the nature of H-bonding on the physical stability of multiple-component systems. The molecular structures of the AAs were plotted in Appendix A.

## 2. Materials and Methods

### 2.1. Materials

Indomethacin (IMC) was purchased from Sigma Aldrich (purity ≥ 99%). It is a model drug which presents a rich and very original through polymorphism through the α–γ monotropic system. Indeed, the commercial and more stable phase, namely the γ form [8], is not the denser phase. The long-range order is induced by the formation of dimers via H-bonding of the carboxylic acid group. The metastable α phase is denser than the γ form because it is composed of more compact trimers [9]. It was shown that devitrification of IMC occurs in the γ or α phase depending on whether the temperature of recrystallization is below or above the T_g_, respectively [10]. The recrystallized α phase remains stable by cooling down to room temperature. The α phase can also be prepared by dissolution in methanol and precipitation in water at room temperature [11]. The molecular structure of IMC and the molecular packing distinctive of the α and γ phases are plotted in Appendix A.

Amino acids (AAs), i.e., L-leucine (LEU, C_6_H_13_NO_2_, purity ≥ 98%), L-norleucine (NLE, C_6_H_13_NO_2_, purity ≥ 98%), L-tert-leucine (TLE, C_6_H_13_NO_2_, purity ≥ 99%) and L-arginine (ARG, C_6_H_13_NO_2_, purity ≥ 98%) were purchased from Sigma Aldrich (St. Louis, MO, USA) and used as received without any further purification.

### 2.2. Methods

#### 2.2.1. Ball Milling

Mixture blends (IMC–AAs) were systematically prepared in equimolar proportions by cryo-milling for 45 min at 30 Hz using Retsch CryoMill. Mixtures of a typical mass of 1 g were milled at −196 °C. The IMC and AAs were placed in a ZrO_2_ jar and milled using one ball (Ø = 20 mm) of the same material. A procedure of alternating milling periods of 5 min with pause periods (milling at 5 Hz) of 1 min was used to limit mechanical heating. After milling, the sample was rapidly prepared at room temperature and cooled down to a low temperature at 6 °C/min for collecting Raman spectra upon heating at 1 °C/min.

#### 2.2.2. Raman Spectroscopy

Raman investigations were performed on two spectrometers depending on the spectral region analyzed. 

One part of these Raman investigations was carried out in the low-frequency region for determining unambiguously the physical state of the multi-component systems obtained by co-milling, as well as their phase transformations. These experiments were performed on the very high-dispersive XY-Dilor spectrometer, composed of three gratings with a focal length of 800 mm. It is equipped with a Cobolt laser emitting at 660 nm. Opening the slits at 150 µm makes it possible to detect a Raman signal down to 5 cm^−^^1^ in high resolution configuration (lower than 1 cm^−^^1^). Milled materials were loaded in spherical Pyrex cells and hermetically sealed. The temperature of the sample was regulated using an Oxford nitrogen flux device that keeps temperature fluctuations within 0.1 °C. Low-frequency Raman spectra were collected between 5 cm^−^^1^ and 200 cm^−^^1^ in 1 min, and they were collected in-situ during heating ramps at 1 °C/min. This experimental condition was systematically used in the methodology developed to determine the T_g_ from low-frequency Raman data. The Raman band shape in the low-frequency region is very sensitive to thermal fluctuations through the Bose factor and requires specific data processing. In the first stage, the Raman intensity ($I_{Raman}\left(\omega ,T\right)$) is transformed into a reduced intensity ($I_{r}\left(\omega \right)$) according to [12,13]:$$I_{r}\left(\omega \right)=\frac{I_{Raman}\left(\omega ,T\right)}{\left[n\left(\omega ,T\right)+1\right]\cdot \omega }$$
where $n\left(\omega ,T\right)$ is the Bose factor. The low-frequency spectrum represented in the reduced intensity is dominated by a very intense component, named quasielastic scattering, detected in the very low-frequency range (<50 cm^−1^) and reflecting local rapid motions [14] or mono molecular motions in very disordered molecular systems such as liquids [15] or plastic crystals [16,17]. This low-frequency contribution is highly temperature dependent, and very sensitive to the ordering/disordering processes responsible for phase transformations [18] such as glass transition and crystallization. The quasielastic scattering generally overlaps with the low-frequency part of the vibrational spectrum. Analyzing both the quasielastic and the pure vibrational spectrum of very disordered states (e.g., the amorphous state) requires the separation of these two contributions in the second stage of the data processing via the fitting procedure described in Figure 1a. 

The quasielastic component is usually described by a Lorentzian band shape centered at ω = 0, and the vibrational contribution of the amorphous states by a log-normal distribution reflecting the vibrational density of states (VDOS) [19]. After removing the quasielastic component, the reduced intensity is converted into Raman susceptibility (*χ*″(*ω*)) according to [20,21]:$$\chi^{\prime \prime }\left(\omega \right)=I_{r}\left(\omega \right)\cdot \omega =\frac{C\left(\omega \right)}{\omega }\cdot G\left(\omega \right)$$

*C(ω)* is the light–vibration coupling coefficient and *G(ω)* is the VDOS, generally obtained from inelastic neutron scattering. In the absence of periodicity conditions, the VDOS reflects the collective vibrations in the amorphous state, roughly corresponding to the spectrum of the lattice modes, as observed in Figure 1b. The integrated intensity of the ω = 0 Lorentzian band is proportional to the integrated intensity calculated in the low-frequency domain (colored in red in Figure 1a) after the spectrum renormalization by the intensity determined in the region colored in blue in Figure 1a, which is almost temperature independent. 

The second part of the Raman investigations was focused on the mid-frequency domain covering the C=O stretching region between 1550 cm^−^^1^ and 1750 cm^−^^1^. Spectra were collected using an InVia Renishaw spectrometer. The 785 nm line emitted from a Fandango Cobolt laser was focused on the powder sample via an achromatic lens for analyzing the largest possible volume of material (about 1 mm^3^). The sample temperature was controlled by placing the sample in a THMS 600 Linkam temperature device. The acquisition time of each spectrum was 1 min, and they were collected in-situ during heating ramps at 1 °C/min. Analyzing the C=O stretching region makes it possible to discriminate the physical state and the polymorphic forms of IMC, characterized by various molecular associations via C=O…H bonding [22,23,24,25,26].

## 3. Results

### 3.1. Analysis of Amorphous IMC upon Heating

In the first step, the amorphous IMC was analyzed upon heating in the low- and mid-frequency regions. The results were considered as references for interpreting experiments performed on IMC–AAs mixture blends.

The low-frequency investigations were performed on amorphous IMC prepared by cryogenic milling and by quenching the liquid state directly at room temperature ($\dot{T}>200\;{}^{\circ}C/\min$), i.e., about 20 °C below the T_g_. The temperature dependence of the quasielastic intensity of amorphous IMC heated at 1 °C/min was plotted in Figure 1c, for amorphous IMC prepared by cryo-milling and by quench of the liquid. The glass transition can be observed via the change in the slope of I_QES_(T) around 40 °C, as was expected from the literature (~42 °C) [10,27]. This signature of the T_g_ in the I_QES_(T) plot was also observed for other molecular materials [18,28,29,30] and interpreted as resulting from the release of motions above the T_g_ enhancing the quasielastic scattering. On the other hand, no glass transition can be detected in the I_QES_(T) plot of the amorphous IMC obtained by cryo-milling, in agreement with previous studies [22,31], since crystallization is detected below the T_g_ via the drop in the quasielastic intensity close to room temperature. We attribute this phenomenon to the fact that surface crystallization [32] resulted from a high specific surface generated by milling. Spectra collected upon heating the cryo-milled powder and the quenched liquid are plotted in Figure 2a and Figure 2b, respectively. Figure 2a clearly reveals that the cryo-milled powder recrystallized in γ form, as expected [10], while Figure 2b shows the change in the low-frequency intensity considered as the signature of the glass transition with detection of the early stage of crystallization via observation of subtle shoulders indicated by two red arrows.

Experiments were performed in the C=O stretching region only on amorphous IMC prepared by quenching the liquid state. The detailed assignment of the Raman bands was previous reported by several authors [23,24,25,26,33] and is given in Appendix A. Special attention has been paid to the 1680 cm^−^^1^ band in the amorphous IMC, attributed to the benzoyl C=O stretching vibrations. The amorphous state and the polymorphic crystalline forms of the IMC can be easily identified from their respective spectra plotted in Figure 3a. The fitting procedure described in Appendix A was used for analyzing the strength of H-bonding in co-amorphous formulations via the position of the broad high-frequency band in the spectrum of the amorphous IMC. The temperature dependence of the band frequency was plotted in Figure 3c. In the low-temperature range, the frequency is almost temperature independent, reflecting the frozen molecular configuration, while above 0 °C a positive T-dependence is observed, usually considered the signature of H-bonding. The temperature behavior of this broad band reflects the properties of the H-bond network of the amorphous IMC, i.e., it is mainly composed of disordered cyclic dimers [26,33].

### 3.2. Co-Amorphous Formulations of IMC with LEU, NLE and TLE

The cryogenic co-milling of the IMC with the three isomeric leucine AAs (LEU, NLE and TLE) produces co-amorphous formulations. The formulations where analyzed upon heating at 1 °C/min both in the low-frequency range and in the C=O stretching region. The spectra collected in the low-frequency region are plotted in Figure 4. This figure shows systematic changes in the very low-frequency region interpreted as resulting from the release of frozen motions, and thus reflecting the glass transition. Additionally, Figure 4a,b clearly shows the presence of phonon peaks above 70 °C, while subtle shoulders (indicated by arrows in Figure 4c) can be identified as corresponding to the most intense phonon peaks of the stable γ phase. The spectra collected at 90 °C are plotted in Figure 5a for easier identification of the recrystallized matter within the IMC–AA formulations. This figure firstly indicates that only the recrystallization of the IMC can be clearly detected. Secondly, the recrystallized state can be unambiguously identified as corresponding to the γ phase for the IMC–LEU and a mixture of the α and γ phases for the IMC–NLE, while it is confirmed that the shoulders detected in the spectrum of the IMC–TLE correspond to the crystallization traces of the γ phase. It is worth noting that the IMC–NLE formulation is poorly crystallized compared with the IMC–LEU formulation. The additional low-frequency intensity detected with respect to the spectrum of the cryo-milled IMC reflects the presence of amorphous matter mostly corresponding to the non-crystallized AAs. The quasielastic intensity in this very low-frequency range is plotted against temperature for the three formulations in Figure 5b. These temperature dependences are compared with those of the amorphous IMC prepared by quenching the liquid and by cryo-milling. The signature of the glass transition, i.e., the change in the slope of I_QES_(T), can be clearly observed in the three IMC–AA formulations at lower temperatures than the T_g_ of the quenched liquid. However, it cannot be concluded that AAs have a plasticizing effect on IMC since cryo-milled IMC recrystallizes close to room temperature. It can be observed that the IMC–NLE, characterized by the lowest T_g_, recrystallizes at a lower temperature than the IMC–LEU. On the other hand, the IMC–LEU and IMC–TLE have similar T_g_ (~30 °C), but the TLE does not recrystallize. 

The C=O stretching region plotted in Figure 3 was investigated in order to analyze the nature of the H-bond interactions. It is worth noting that the Raman signals of the co-amorphous formulations mostly reflect the behavior of the IMC, since the Raman signals of AAs are not perceptible compared with that of IMC, as is shown for the crystalline compounds in Appendix A. The Raman signal in the APIs is generally strongly enhanced with respect to that scattered by excipients because of the presence of π bonds within the API molecules [12,13]. Given that the broad band at 1680 cm^−^^1^ exhibits the signature of H-bonding interactions between IMC molecules, i.e., a positive temperature dependence of the band position (Figure 3c), this temperature dependence was plotted in Figure 6a for the three IMC–AA formulations and compared with that of the cryo-milled IMC. In the low-temperature range (T < 0 °C), the band position in the glassy IMC is almost temperature independent, while above 0 °C the band shifts significantly toward the high frequencies upon heating, which is the signature of H-bonds involving C=O groups. In the low-temperature range, it is also noticeable that the band is drastically shifted toward the low frequencies in the IMC–AA formulations, reflecting the strengthened H-bond interactions in presence of the AAs. Upon heating, the shift of the band in the IMC–AA formulations compared with the pure IMC reduces and vanishes for the IMC–LEU and IMC–NLE formulations in which crystallization was detected. In the IMC–TLE formulation, the band is still downshifted with respect to the pure IMC. This could indicate that the stabilization of the H-bond network of the amorphous IMC inhibits crystallization. Figure 6b confirms the absence of crystallization in IMC–TLE formulation, the partial recrystallization of the IMC in the γ form in the IMC–LEU, and the recrystallization of the IMC mostly in the α form, with a subtle trace of the γ form highlighted by the arrow. Additionally, a high degree of crystallization of the IMC in the IMC–NLE can be observed compared with that in the IMC–LEU.

### 3.3. Co-Amorphous IMC–ARG Formulation

The spectra collected in the low-frequency and C=O stretching regions upon heating the IMC–ARG formulation are plotted in Figure 7a and Figure 7b, respectively. Both figures clearly show that the amorphous IMC can be stabilized up to 100 °C without the detection of any trace of crystallization. Figure 7a indicates a change in the temperature dependence of the low-frequency intensity, indicated by the arrow and interpreted as the signature of the T_g_. This change can be better visualized by the plot of the temperature dependence of the quasielastic intensity in Figure 8a. It can be compared with the I_QES_(T) determined in the glassy IMC and in the co-amorphous IMC–TLE mixture. It can be observed that the T_g_ of the IMC–ARG (~80 °C) was higher than the T_g_ of the pure IMC. The differential scanning calorimetry data obtained by the calorimetric experiments shown in Appendix A confirmed the high T_g_ value of the IMC–ARG co-amorphous formulation. The quasielastic intensity is also plotted (in Figure 8b) vs. the temperature for the ARG amorphized by cryo-milling and the crystalline ARG. This reveals that the T_g_ of the ARG was about 55 °C, which is in agreement with a previous determination from freeze-dried sample [34]. Consequently, the high T_g_ value of the IMC–ARG with respect to the IMC cannot be interpreted as an antiplasticizing effect of ARG. Such an unusually high T_g_ value was previously determined in the IMC–ARG co-amorphous formulations prepared by ball milling and spray-drying [5]. This phenomenon was explained by the salt formation revealed from very clear spectral changes in the C=O stretching region, also investigated in the present study. It is worth noting that salt formation has been systematically detected by significant spectral modifications in this C=O stretching region [5,35]. Figure 7b clearly shows no spectral modification of the C=O stretching spectrum, as would be expected for salt formation.

The temperature dependence of the C=O stretching band located around 1680 cm^−^^1^ was analyzed, especially its position plotted against the temperature in Figure 9a. This plot mimics that obtained from the analysis of the IMC–TLE co-amorphous mixture, but significantly downshifted. Spectra collected at low temperature in the IMC–TLE and IMC–ARG plotted in Figure 9b show that only this band is shifted in this spectral region, reflecting a strongly enhanced effect of the ARG on the H-bonding properties of the IMC with respect to the other AAs. Intriguingly, a change in the ω(T) curves can be systematically observed around 40 °C, i.e., around the T_g_ of the IMC, in Figure 9a as well as in Figure 6b. This feature could be interpreted as a signature of the change in the molecular mobility in the IMC, despite the formulation exhibiting a single-phase behavior. 

### 3.4. Hydration Stability

The hydration stability of the IMC–TLE, IMC–LEU and IMC–ARG co-amorphous blends was analyzed by exposing prepared powders to 98% RH in desiccators containing a saturated solution of potassium sulfate. Raman spectra were collected at various exposing times, and these are plotted in Figure 10. As could be expected from the analysis of the thermal stability of the IMC–AA blends, Figure 10c highlights the high stability of the amorphous IMC within the IMC–ARG blend compared with other blends prepared with LEU and TLE. However, it can be observed that the IMC–TLE is slightly less stable than the IMC–LEU under 98% RH, while it can be maintained in a co-amorphous state at higher temperatures than the IMC–LEU.

## 4. Discussion

This study confirms the high capability of AAs to stabilize drugs in an amorphous state within co-amorphous blends obtained by cryo-milling. Combining low- and mid-frequency Raman investigations carried out in-situ upon heating co-amorphous blends over a wide temperature range (from −100 to 100 °C) has provided important information on the mechanism by which the drug is stabilized and the degree to which the AAs effectively stabilize the drug in its amorphous state. Selecting indomethacin, characterized by a rich polymorphism related to various hydrogen bond networks, as a model drug made possible the analysis of the influence of the AAs on the H-bonding associations of the IMC molecules. Using isomeric leucine molecules as co-formers in the preparation of formulations by cryo-milling enabled the stabilizing of the IMC at room temperature, while the amorphous cryo-milled IMC was unstable at room temperature and recrystallized into the γ phase. The IMC–AA amorphous blends were more stable than the glassy IMC obtained by quenching the liquid state below $T_{g}\approx 40\;{}^{\circ}C$. The glass transitions in the blends prepared with LEU, NLE and TLE were observed at temperatures lower than 40 °C, and IMC recrystallization was observed in all except the IMC–TLE blend. It is worth noting that only the recrystallization of the IMC was clearly detected in crystalline states identified in the low- and mid-frequency regions. It is also worth noting that the recrystallization occurs in different states with different degrees of crystallization in the IMC–LEU and IMC–NLE formulations. The recrystallization of these two formulations can be explained by the thermal activation of local motions in the LEU and NLE generating instabilities in the crystalline lattice of these two amino acids [36,37], an observation not reported in the literature for TLE. This study also shows that the three isomeric leucine molecules have a similar influence on the 1680 cm^−1^ band position distinctive of the H-bonding associations of IMC molecules in the low-temperature range. The downshift of the band in the co-amorphous blends is the signature of H-bonding strengthening. At the high temperatures, the band was still observed at lower frequencies only in the IMC–TLE. This indicates that the more important capability of TLE is to stiffen the H-bonds of the IMC within the co-amorphous formulation, inhibiting the recrystallization of the IMC. This demonstrates that the intrinsic properties of AAs significantly impact the stability of amorphous IMC. It is likely that the molecular conformation and the hydrophobic character favor the local disordering of side-chains and thus have a negative effect on the co-amorphous stability. It can be concluded that the low hydration stability of the amorphous IMC in formulations prepared with leucine isomers may be related to the hydrophobic character of the AAs.

This study has revealed the capability of ARG to maintain amorphous IMC within blends exposed to high temperatures or high RHs. The single-phase behavior of the IMC–ARG co-amorphous formulation is characterized by the intriguing high temperature of the glass transition, likely related to the strengthened H-bonding network in the IMC inducing the stiffening of the glassy matrix. The special impact of the ARG on the stability of the co-amorphous blend with respect to the leucine isomers can be explained by the absence of local disorder (i.e., the absence of methyl groups) and its capability to strengthen the H-bond network of the IMC within the co-amorphous blend because of its hydrophilic character. 

## 5. Conclusions

The present study points out the need to analyze the thermal and hydration stabilities of co-amorphous blends for selecting a co-former in order to design co-amorphous formulations providing maximum stability of the amorphous API. It includes a necessary and complementary analysis of a screening method for selecting the best co-former for designing co-amorphous formulations. The glass transition temperatures of co-amorphous mixture blends were determined and the properties of the H-bonds in amorphous indomethacin were analyzed. Additionally, this study has highlighted the exceptional capability of arginine for stabilizing amorphous indomethacin, a finding which is in agreement with a previous study of spray-dried co-amorphous formulations [5] in which this phenomenon was explained by salt formation detected in the C=O stretching vibrations. The absence of a Raman signature for salt formation in the present study motivates deeper analyses of the IMC–ARG mixtures that could be a promising route for stabilizing amorphous APIs. 

## Figures and Tables

**Figure 1 pharmaceutics-15-00337-f001:**
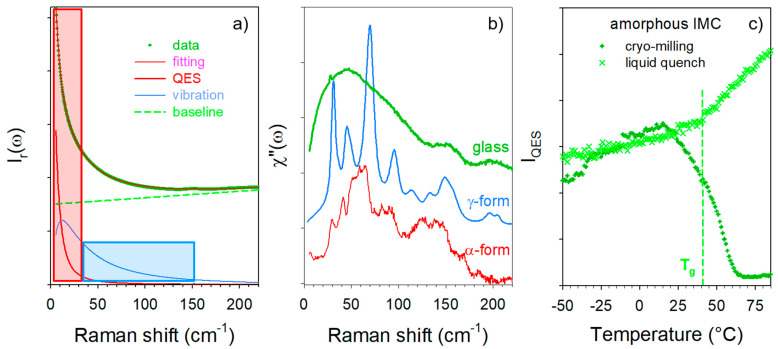
Low-frequency analysis of amorphous indomethacin. (**a**) Description of the fitting procedure of the reduced intensity (Ir(ω)) spectrum used for separating quasielastic scattering from the vibrational spectrum. The blue area was used for the spectrum renormalization while the red area is proportional to the quasielastic intensity. (**b**) Representation of the vibrational spectrum in Raman susceptibility very close to the VDOS of glassy IMC compared with the lattice mode spectra of α and γ polymorphs at room temperature. (**c**) Temperature dependence of the quasielastic intensity calculated in amorphous IMC prepared by cryo-milling and by quenching the liquid.

**Figure 2 pharmaceutics-15-00337-f002:**
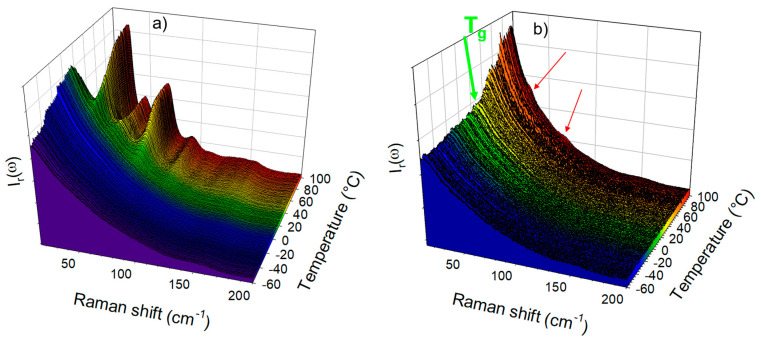
Low-frequency spectra collected in-situ along the heating ramp at 1°C/min of amorphous IMC prepared (**a**) by cryo-milling and (**b**) by quenching the liquid.

**Figure 3 pharmaceutics-15-00337-f003:**
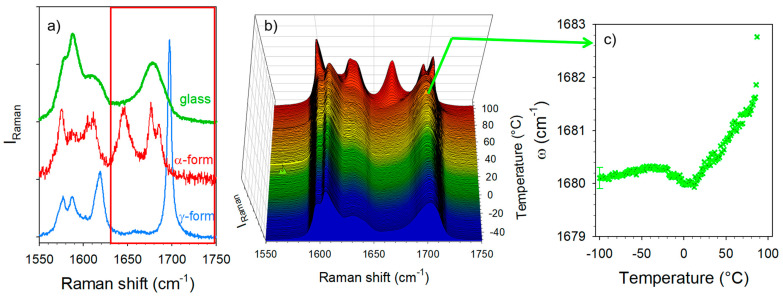
Analysis of the C=O stretching spectrum of IMC. (**a**) The spectrum is plotted in the glassy state and in α and γ polymorphs, and the spectral range in the red frame has been analyzed in this study; (**b**) temperature dependence of the C=O stretching spectrum; (**c**) temperature dependence of the Raman band which exhibits the signature of H-bonding.

**Figure 4 pharmaceutics-15-00337-f004:**
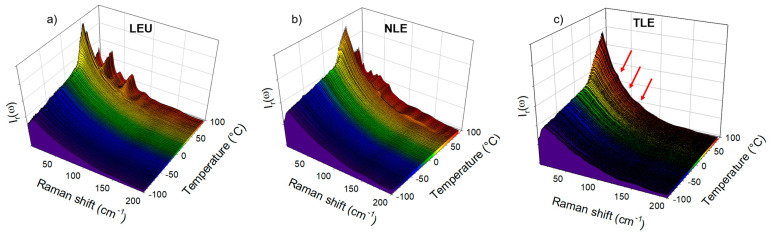
Temperature dependence of the low-frequency spectrum of the IMC–AA blends. Spectra were collected upon heating at 1 °C/min. (**a**) IMC–LEU; (**b**) IMC–NLE; (**c**) IMC–TLE.

**Figure 5 pharmaceutics-15-00337-f005:**
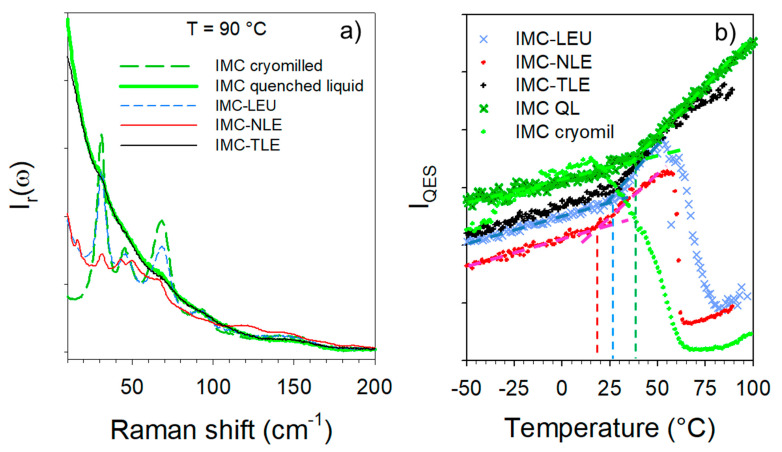
Analysis of the LFRS of the IMC–AA blends. (**a**) Spectra collected at 90 °C; (**b**) temperature dependence of the QES.

**Figure 6 pharmaceutics-15-00337-f006:**
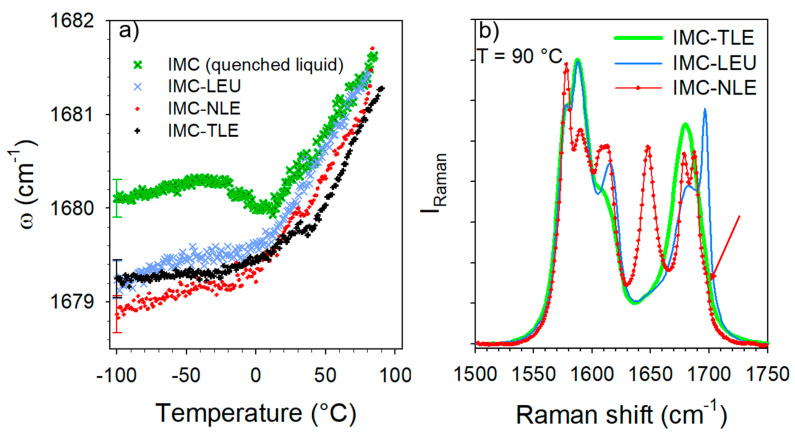
Analysis of the C=O stretching region of the IMC–AA blends. (**a**) Temperature dependence of the band frequency distinctive of H-bonding—only one error bar was reported for better clarity. (**b**) Spectra of the IMC–AA blends collected at 90 °C. The red arrow showing the subtle shoulder reflecting a very weak proportion of the γ phase after recrystallization mostly in the α phase in the IMC–NLE.

**Figure 7 pharmaceutics-15-00337-f007:**
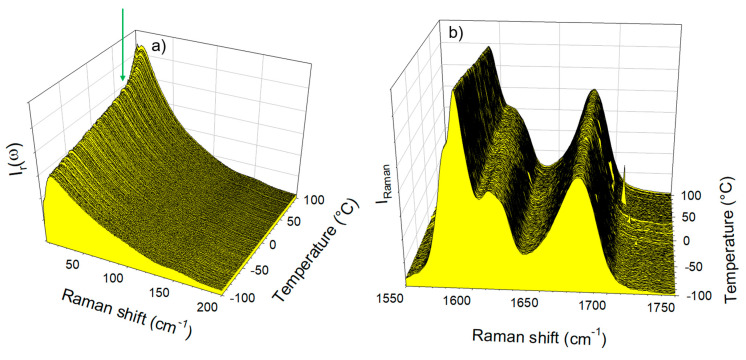
Temperature dependence of the Raman spectrum of the IMC–ARG blend collected upon heating at 1 °C/min (**a**) in the low-frequency region (the arrow indicating the T_g_) and (**b**) in the C=O stretching region.

**Figure 8 pharmaceutics-15-00337-f008:**
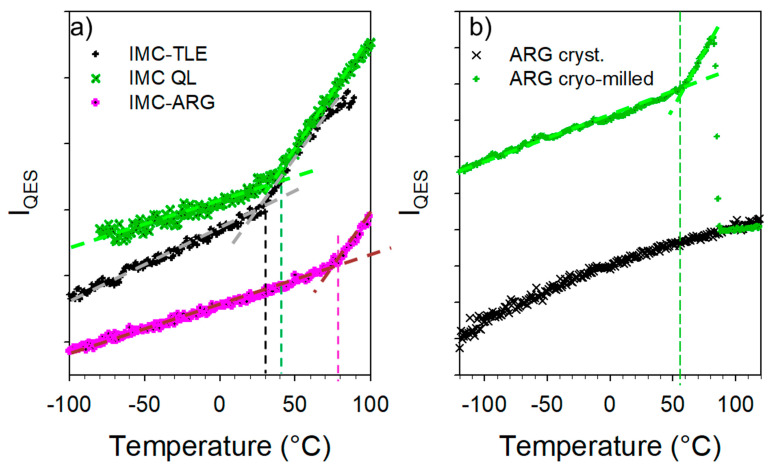
Temperature dependence of the quasielastic intensity (**a**) determined for IMC–TLE and IMC–ARG blends compared with that of the glassy state of the IMC and (**b**) determined in the amorphous ARG compared with that of the crystalline ARG.

**Figure 9 pharmaceutics-15-00337-f009:**
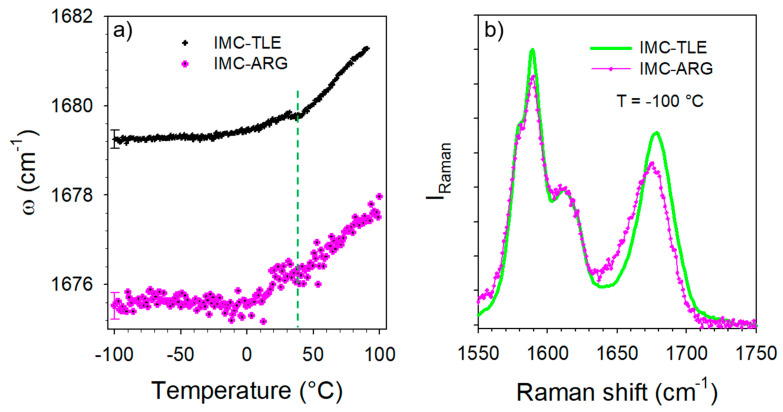
Temperature dependence of the band distinctive of the H-bonding in the IMC–TLE and IMC–ARG blends. (**a**) T-dependence of the frequency—only one error bar was reported for better clarity. (**b**) Spectrum collected at −100 °C in the IMC–TLE and IMC–ARG blends.

**Figure 10 pharmaceutics-15-00337-f010:**
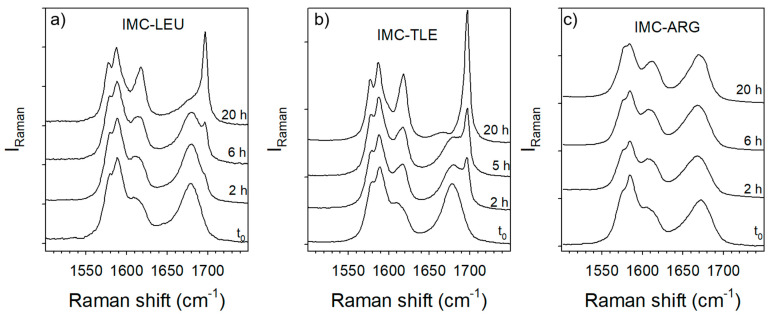
Time dependence of the C=O stretching spectra of the co-amorphous blends exposed to 98% RH: (**a**) IMC–LEU; (**b**) IMC–TLE; (**c**) IMC–ARG.

## Data Availability

Not applicable.

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
