# Peer review of "Mechanism for Stabilizing an Amorphous Drug Using Amino Acids within Co-Amorphous Blends"

_pharmaceutics, 2023, doi:10.3390/pharmaceutics15020337_

Round 1
Reviewer 1 Report
The Authors of manuscript 2111669 report on the characterization by Raman spectroscopy of amorphous indomethacin, and of binary mixtures of indomethacin with selected amino acids.
The manuscript lacks important information, and it seems that it did not undergo a thorough revision by the Authors before submission. I also think that the characterization and analysis of the binary systems, and especially of the indomethacin-arginine system, is incomplete (points 5 and 6 below).
I would advise the Authors to resubmit an updated version of their manuscript that includes further characterization of the behaviour of Tg with composition in the indomethacin-arginine mixtures, and a more thorough analysis of the C=O bands of amorphous indomethacin and amorphous mixtures.
Here are my detailed comments:
(1) In the introduction, the Authors state that their aim is “to analyze multi-component systems” (line 41), but they really only study binary systems. Moreover, in the same paragraph they state that “Indomethacin (IMC) and ibuprofen (IBP) were selected as model drugs from consideration of Tg values, significantly different (~ 40 °C and -50 °C for IMC and IBP respectively)” (Lines 45-47), while in reality they only measure indomethacin samples which do not contain any ibuprofen.
(2) The following information about the experimental procedure is missing in the Methods Section:
- what cooling rate was employed to obtain the glassy samples by supercooling?
- If the commercial indomethacin sample was initially in the gamma phase (line 57), and “the cryo-milled powder recrystallized in γ form” (line 134) how then was the alpha phase obtained? (data shown in Figures 1 and 3)
- were the Raman spectra taken isothermally at fixed T, or did the temperature increase during acquisition of the spectra? The sentence “Low-frequency Raman spectra have been collected between 5 and 200 cm-1 in 1 minute, in-situ during heating ramp at 1 °C/min.” suggests that T was not fixed for each spectrum. If this is indeed so, why were the measurements performed this way?
(3) In Figure 1c, as well as in both panels of Figure 8, linear fits should be shown of the high and low T portions of the data of the liquid-quenched indomethacin in Fig. 1c, and of milled samples in Fig. 8, and the determination of the Tg value as intersection of these two linear fits should be shown explicitly. Also, experimental values of Tg for indomethacin and arginine reported in previous studies should be given in the text.
Can the Authors rationalize why the milling procedure was more successful for arginine than for indomethacin (one crystallized below Tg while the supercooled liquid of the some compound, and the other cryomilled mixture, did not) ?
(4) the composition of the binary mixtures, in terms of the molar fraction of the amino acid and the drug, is never stated in the manuscript. This information is obviously relevant for other researcher to reproduce the manuscript results, besides for understanding the meaningfulness of the reported changes in Tg and Raman peaks.
(5) The Authors find that the Tg of the arginine-indomethacin mixture is significantly higher than those of either separate amorphous component. This is an interesting and uncommon result, which should be further corroborated and investigated. The Authors should perform a study as a function of relative concentration of arginine, to see whether this behaviour is common to several concentrations and whether the Tg follows a smooth dependence on arginine content.
(6) what are the different bands in the amorphous phase of indomethacin in Figure 3a? Since the Authors only discuss in detail these Raman features of all the available spectrum, it is fair to ask them to provide an assignment. Indomethacin contains for example two non-equivalent carbonyl groups. Understanding the origin of the spectral features would help understand the differences observed between the various samples in Figures 6b and 9b. Also amino acids contain carbonyl groups, where do they contribute to the observed Raman spectrum?
The Authors could also mention what assignment has been previously provided for the crystalline (alpha and gamma) phases, if available.
I add a couple of comments concerning the format of the manuscript:
(7) the manuscript contains an exceptionally long Methods section, which actually contains what can be considered proper results of the work. I recommend restructuring the paper, placing at least figures 2 and 3 in the Results section. Also, the manuscript is relatively short and I do not see the need for a supplementary information file. I think that the figures shown there should be incorporated in the main article, adding also the molecular structure of indomethacin (these new figures can be inserted either as separate figures or as panels to the existing ones).
(8) the sentence “Figure 2b shows the change in the low-frequency intensity considered as the signature of the glass transition with detection of the early stage of crystallization, via observation of subtle shoulders localized by arrows.” (lines 135-137) is misleading. The only arrow in Fig. 2b marks the Tg. Also, are the Authors sure that the “subtle shoulders” are not simply due to experimental noise, rather than crystallization? Even if there were systematic changes ascribable to nucleation of the crystal phase, they should be pointed out in Figure 1 c, where the low-frequency signal is plotted in detail.
Minor points:
- Three different abbreviations are used for “amino acids”, namely AAs, AAS, and A As . Please use only one
- in the abbreviation for the glass transition temperature (Tg), the letter “g” should appear as subscript
- the words “dimer” and “dimers” are misspelled as “dimmer” and “dimmers”
Reviewer 2 Report
The Authors studied the physical stabilization of indomethacin in mixture with some aminoacids (L-leucine, L-norleucine, L-tert-leucine and L-arginine), prepared by cryo-milling, by low-frequency and high-frequency Raman measurements achieving information about the mechanism by which the active principle is stabilized in the amorphous phase.
This is obviously a very important topic for the pharmaceutical researchers and therefore certainly of interest to the readers of this journal.
The manuscript is written very well. The text is flowing and accompanies the reader very well to understand the behavior of the systems and the reasoning of the experimental data.
However, some changes are required:
-Introduction Section, pag.1, lines 35-40: it is necessary to insert some bibliography references to which the Authors are referring.
-In the text, the references to the figures in the Supplementary Material are missing.
-Pag. 5, line 156: … the polymorphic for of IMC… , please correct to … the polymorphic forms of IMC…
-Pag. 8, line 235: change “figure 7b” to “figure 8b”.
-The Authors could expand the paragraph of the Conclusionsin order to better highlight the importance of their study.
-The present work would acquire greater significance if the Raman investigations were supported by Differential Scanning Calorimetry measurements in confirming the glass transition temperature, polymorphic phase etc.
Round 2
Reviewer 1 Report
The Authors complied only partially with the issues raised, and they failed to answer adequately in the case of what I had indicated as the weakest points of their paper, or answered without making any change to their manuscript. I think I am being reasonable, and I only try to contribute to make the manuscript more sound, more correct and more complete.
In the following I use the numbering of my first review.
(2) To my question “ what cooling rate was employed to obtain the glassy samples by supercooling?”, the Authors replied “For the glassy state of pure IMC, the liquid state was quenched at room temperature, and the cooling rate can’t be measured in this case.”
I am well aware that the Authors do not supercool the ball-milled samples. I reformulate my concern: in the manuscript method section, there is only a section on ball milling, and no mention of the supercooling procedure. This information must be added, even if it was employed only for pure IMC. The Authours must have at least an idea of the cooling rate. If it was not constant, it will enough to state that the “cooling rate was of the order of…” or “higher than…”
To my question as to why the Raman spectra were acquired by continuously varying the temperature, their answer was that “This is an experimental condition systematically used for determining the glass transition temperature from low-frequency Raman data. This condition was defined to mimic experimental conditions used to determine Tg from DSC data. This condition was systematically used in Ref. 13, 18,-20 for determining Tg in other molecular materials”
This answer is convincing, so why don’t the Authors include this information in the manuscript?
(3) to my comment that linear fits should be used to determine the Tg’s, the reply was that “The main goal of this study was not determining accurately Tg values” and that they do not do linear fits because “If we perform the linear fits of data in 2 separate low- and high-T data regions, it is necessary to arbitrarily separate these regions undoubtedly at the temperature corresponding to the vertical dashed lines.”
I disagree with the statement about the main goal of this study, which according to the Authors is “the manuscript focuses on the ability of amino acids to stabilize amorphous [indomethacin]”. As a matter of fact, this ability of aminoacids was already proven in a more systematic study that included especially Arginine, in a previous article appeared in this same journal, namely in Pharmaceutics 10, 47 (2018) (see also my point (5) below). Therefore, also “details” such as the Tg of their amorphous samples count, for their article; it is not enough to state that their goal is different.
And there are standard procedures to decide which points belong to the low-T or high-T region in a fit, a simple solution being not to consider “problematic points” and simply fit linearly well separated groups of points. Any linear fit, regardless of which points are considered, would definitely be much sounder than the “guides to the eye” that the Authors show only in one case, and only in the supplementary information file (Fig. S5). Instead, the same procedure should be done for all Tg determinations, and shown explicitly. (I would also recommend the Authors to show such linear fits choosing other markers than crosses; circles are better).
(5) I thought that the unexpected result for the Tg of the binary amorphous mixture Arginine+indomethacin obtained by ball-milling was new, but it turns out that exactly the same thing was already published in Pharmaceutics 10, 47 (2018) by the group of Thomas Rades. These researchers ascribe the much higher Tg of indomethacin mixtures with aminoacids to salt formation, that is to charge transfer leading to much stronger intermolecular interactions. The result of the present manuscript are in line with those of Rades and coworkers. This should be stated and duly acknowledged, and the possibility that salt formation is the cause of the increased Tg, as already found for other aminoacids, should also be stated clearly. The Authors should also compare their Tg with that of the equimolar mixture of Pharmaceutics 10, 47 (2018). I suggest also to remove the newly added sentence in the conclusions, that “This feature [unusually high Tg] will be carefully analyzed very soon.”, and to analyze the possibility that (at least partial) salt formation is the reason behind the observed stability of the amorphous binary mixture.
Finally, in answer to my concern about the reliability of Tg values obtained by Raman, the Authors have added a new figure S5 in the supplementary information, where in the upper panel they show a DSC trace and claim that it gives the same Tg as Raman, namely around 80 ºC.
Regretfully, their analysis of the DSC trace is unclear or wrong. Tg values in DSC are obtained from the low-temperature onset in heat-up scans. If the data they show is a heat-up scan, then the correct determination of Tg is around 65 ºC, and not 80 ºC, which is instead the high-temperature offset. It might be that the actual Tg depends on the degree of salt formation, and that this might be different in different samples (the Tg obtained by Rades and coworkers. is for example higher than the one reported here). Whatever the cause of the different Tg, the upper panel of Figure S5 must be modified (or removed, if the Authors prefer to base their discussion only in comparison with the results of the Rades group).
(6) I had asked the Authors to provide an assignment of Raman features of their amorphous indomethacin samples. The Answer was “The assignment of Raman bands was already performed by several authors and was referenced in the 1st version of the manuscript 21 to 25. The assignment of the broad Raman band around 1680 cm1 which exhibits the signature of H-bonding was added with an associated reference (Hédoux et al. Phys. Rev. B 77, 094205, 2008) in page 6 between lines 174 and 188. In this new reference, the detailed assignment of Raman bands detected in this spectral region has been given in Table 1.”
Different authors have studied the crystalline forms, and these are not of interest here. In the manuscript the Authors only added a sentence about the band at 1680 cm-1, what about the other bands? If they are the same as in the crystalline phases, or in mixtures with PVP, or in indomethacin amorphized with a different method, this is also a result worth being stated in the manuscript, and the corresponding assignment should be given in this manuscript.
I really do not understand why a reader of this manuscript should be forced to look up table 1 in PRB 77, 094205, or Table 1 in Pharm. Res. 14, 1691 , to know what the measured Raman bands correspond to…
Concerning the issue of the contribution of aminoacids to the Raman spectra, I do not think that it is enough to show that their crystalline form does not yield a strong Raman signal, what about their amorphous form? If it is known in the Raman community that, for some reason, this particular spectral region is not very Raman-active in aminoacids in amorphous form, this should be stated and properly referenced.
Author Response
Responses to Reviewer 1 were uploaded
No additional revision was requested by Reviewer 2

Reviewer 2 Report
The Authors made the suggested changes.
Author Response
no additional revision was requested by Reviewer 2